# Disparities in Cancer Incidence across Income Levels in South Korea

**DOI:** 10.3390/cancers15245898

**Published:** 2023-12-18

**Authors:** Su-Min Jeong, Kyu-Won Jung, Juwon Park, Nayeon Kim, Dong Wook Shin, Mina Suh

**Affiliations:** 1Department of Medicine, Seoul National University College of Medicine, Seoul 03080, Republic of Korea; smjeong.fm@snu.ac.kr; 2Department of Family Medicine, Seoul National University Hospital, Seoul 03080, Republic of Korea; 3National Cancer Control Institute, National Cancer Center, Goyang 10408, Republic of Korea; ara@ncc.re.kr (K.-W.J.); jwpark7@ncc.re.kr (J.P.); fatima@ncc.re.kr (N.K.); 4Department of Clinical Research Design and Evaluation, Samsung Advanced Institute for Health Science and Technology, School of Medicine, Sungkyunkwan University, Seoul 16419, Republic of Korea; 5Department of Family Medicine and Supportive Care Center, Samsung Medical Center, Seoul 06351, Republic of Korea

**Keywords:** income, colorectal cancers, incidence rate, Korea, thyroid cancer

## Abstract

**Simple Summary:**

In this research, we aimed to determine how different income levels relate to the incidence of cancer and the stage of cancer at diagnosis in South Korea. Using Korea Central Cancer Registry data from 2018, we found that the low-income group had higher rates of certain cancers, like colorectal and cervical cancers, while the high-income group had higher rates of some cancers, like thyroid and prostate cancers. Overall, low-income groups were more likely to be diagnosed with advanced-stage cancer. These results highlight the importance of understanding existing disparities in cancer incidence and the stage at cancer diagnosis among income levels and the importance of developing strategies to mitigate these disparities.

**Abstract:**

Background: Recent nationwide studies of disparities in cancer incidence by income are scarce in Korea. This study investigated such disparities in cancer incidence and the stage at cancer diagnosis across income groups in Korea. Methods: This study utilized data from a national cancer database, specifically focusing on cases recorded in the year 2018. Income levels were categorized into quintiles according to the insurance premium paid in addition to the Medicaid benefit. The slope index of inequality (SII) and relative index of inequality (RII) were used to measure absolute and relative differences in cancer incidence by income. A multivariable logistic regression was performed to estimate the risk of a distant stage at cancer diagnosis. Results: The total number of cases of incident cancer was 223,371 (men: 116,320, women: 107,051) with shares of the total of 29.5% (5Q), 20.4% (4Q), 16.0% (3Q), 13.5% (2Q), 15.6% (1Q), and 5% (Medicaid). The most common cancer type was thyroid cancer, followed by gastric and colorectal cancers. The age-standardized incidence rate for all cancers was lowest in the highest income group, but the SII was not statistically significant (SII: −35.7), and the RII was −0.07. Colorectal and cervical cancers had lower incidence rates for higher income groups, while thyroid and prostate cancers had higher incidence rates for higher income groups. The odds ratio for a distant stage at diagnosis for all cancers increased for lower income groups relative to 5Q. Conclusions: Disparities in cancer incidence in a Korean population differed by cancer type, and lower income was a significant predictor of a distant stage at diagnosis for cancers overall. These results emphasize the need for further study of the underlying causes of disparities in cancer incidence and the stage at diagnosis, as well as the need for interventions to mitigate these disparities.

## 1. Introduction

The burden of cancer incidence and mortality is rapidly growing worldwide due to aging populations and changes in the prevalence of cancer risk factors related to socioeconomic development [1]. In 2020, it is estimated that there were 19.3 million incident cancer cases and 10 million deaths from cancer globally [1]. In Korea, the rate of cancer incidence was 275.4 per 100,000 in the population (age-standardized rate) in 2019, showing a stable trend since 2015 [2].

Disparities in cancer incidence result from health-related behaviors, such as diet quality, physical activity, smoking, and alcohol consumption [3], as well as health system factors, including the availability of cancer screening programs and health care access. In addition, disparities in cancer incidence differ by cancer type [4]. For example, previous studies drawing on the 2009 data from the Korean national health insurance service (K-NHIS) linked to a cancer registry reported that stomach, lung, and rectal cancer incidence risks increase as income levels fall, while the risks of thyroid and prostate cancer in men increase with income levels. In a Canadian study, individuals with low incomes had higher cancer incidences for head and neck, oropharyngeal, stomach, colorectal, liver, lung, cervical, and kidney cancers [5]. Conversely, individuals with higher incomes had higher cancer incidences for melanoma; leukemia; lymphoma; and breast, uterine, prostate, and testis cancers.

Socioeconomic status is a significant predictor of the stage at cancer diagnosis [6]. In the US, data reported by the Surveillance, Epidemiology, and End Results (SEER) program indicate that a late-stage diagnosis is associated with lower socioeconomic status due to health care accessibility, particularly cancer screening [6]. In addition, higher county poverty was significantly associated with a distant stage at cancer diagnosis in the US [7]. A recent Korean study found that the incidence of cervical cancer was greater in the highest quintile than in lower income quintiles, while the risk of distant-stage cervical cancer at diagnosis was greater among women with lower income levels (odds ratio (OR): 1.46; 95% CI: 1.24–1.82), along with a higher 5-year mortality following cancer diagnosis [8]. Consistent with this, breast cancer incidence is higher among women with higher income levels, while a higher risk of distant metastasis at presentation is associated with a lower income level [9].

In spite of these abundant previous studies, updated nationwide studies in the Korean context remain scarce. Previous studies in Korea utilized data from 2001 and 2009 [4,10]. Furthermore, earlier studies had a few limitations, including county-level and residential-area-based income comparisons [6,7,11], not comparing household or individual incomes, using a self-reported income assessment [6], not adjusting for cancer-related risk factors (e.g., smoking and alcohol consumption) [4,6,9], and focusing on a specific type of cancer [6,8,9]. Our study, in contrast, used the latest nationwide data on cancers overall. Our study investigated disparities in cancer incidence and the stage at diagnosis using individual-level incomes and included common cancer types.

## 2. Materials and Methods

### 2.1. Data Source and Study Participants

This study used a national cancer database of data recorded in 2018, which was established by linking the Korea Central Cancer Registry (KCCR) and the K-NHIS claims database. The KCCR data are from a population-based cancer registry established in 2002, including sex, diagnosed cancer type (according to the International Classification of Diseases, 10th Revision), age at diagnosis, cancer stage (SEER), and the date of cancer diagnosis. Every year, the KCCR gathers information on patients diagnosed with cancer at hospitals throughout Korea over the preceding year. K-NHIS is a single government insurer that provides mandatory universal coverage for 97% of the Korean population, where the lowest 3% by income are covered by the National Medical Aid (NMA) program. K-NHIS data include demographic information, socioeconomic status (income, employment status, and residential area), and information from the biennial national health checkup. This linkage database covers all Korean citizens, as the universal national insurance system is provided by a single national insurer (K-NHIS). Since 2004, Korea has been implementing a National Cancer Screening Program (NCSP) for five common cancers (stomach, colorectal, cervical, breast, and liver cancer) that is free to the consumer, with the aim of improving cancer survival. The participation rate of the NCSP was 49.2% in 2020 [12].

We initially included 223,371 participants who had received a cancer diagnosis in 2018 in our analyses of income disparities; after excluding those with missing covariate-related data, 201,096 participants were included in the logistic regression analyses for a distant stage at diagnosis by income.

This study was approved by the Institutional Review Board of the National Cancer Center (IRB No. NCC 2021-0264).

### 2.2. Identification of Cancer Incidence

The study participants were those patients diagnosed with the primary cancer codes C16 (stomach cancer), C18–C20 (colorectal cancer), C22 (liver cancer), C33–C34 (lung cancer), C50 (breast cancer), C53 (cervical cancer), C61 (prostate cancer), C73 (thyroid cancer), or C00–C96 (all cancers) in 2018, using the ICD-10 codes. The total number of cancer patients was 223,371.

The cancer stages included localized, regional, distant, and unknown and were assessed using the information obtained from the SEER program of the National Cancer Institute. A localized cancer was defined as a neoplasm that was entirely confined to an organ, without serosal involvement. A regional cancer was a neoplasm that extended beyond the limits of the organ, invading the surrounding tissues. Distant cancer was a neoplasm that was spreading to parts of the body that were remote from the primary tumor. An unknown stage was defined as a neoplasm with information that was unavailable or insufficient to assign a stage. To evaluate the associations among the income levels and the risk of a distant stage, we combined the “localized” and “regional” stages into the single category “loco-regional” vs. a distant stage.

### 2.3. Income Classification

First, the income level was broadly classified into NMA beneficiaries and NHIS subscribers, where NMA beneficiaries are 3% of the Korean population, who are living under the national poverty line or without an identifiable source of income. The NMA program is a form of subsidized government assistance that is used to provide coverage for healthcare services to the low-income population. The remainder of the population consists of NHIS subscribers, who pay monthly insurance premiums. The insurance premium is calculated based on the monthly salary for employees and the household income, assets, vehicle ownership, and the ages of household members for self-employed individuals and can be used as a proxy for income.

NHIS subscribers were subclassified into five segments according to insurance premium deciles and were further categorized into quintiles, including the lowest segment (insurance premiums 1–4, 1Q) and the highest segment (insurance premiums 17–20, 5Q). The three remaining segments were grouped as 2Q (second quintile, insurance premiums 5–8), 3Q (third quintile, insurance premiums 9–12), and 4Q (fourth quintile, insurance premiums 13–16).

### 2.4. Covariates

The sociodemographic characteristics included sex, age, residence, and employment type. Residences were divided across 17 metropolitan cities and provinces, known collectively as Sido (administrative units). Employment status (or insurance type) was categorized into three groups: NMA beneficiaries, employee-insured, and self-employed insured. The body mass index (BMI, kg/$m^{2}$) was calculated by dividing the body weight in kilograms by the height in meters squared. Information on smoking status and alcohol consumption was collected from a self-report questionnaire. Smoking status was grouped into non-smokers (persons who had never smoked or had smoked less than 100 cigarettes in their lifetimes), past smokers (persons who had smoked in the past but had quit smoking), and current smokers (persons who currently smoke). Alcohol consumption was stratified into an alcohol consumption group and a no alcohol consumption group.

### 2.5. Analytical Approach and Statistics

In this study, a frequency analysis was used to compare baseline characteristics across cancer types, as shown in Table 1. The dependent variable was the cancer incidence rate. Rates were age-standardized by incorporating the corresponding data for population size and age distribution from 2020. Two composite indices of disparity (the slope index of inequality (SII) and relative index of inequality (RII)) were used to measure absolute and relative differences in cancer incidence by income [13]. The SII was calculated as the predicted absolute difference in cancer incidence between the lowest income group and the highest income group. The RII was calculated as the SII divided by the average population health.

We computed adjusted odds ratios for each cancer, using a multivariable logistic regression to estimate the risk of reaching the distant stage at cancer diagnosis, excluding the “unknown” stage. The covariates included sex, age, employment type (whether an employee or a local subscriber), residence (17 Sido), BMI, smoking status (never, past, or current smokers), and alcohol consumption (yes or no). The total population included 201,096 individuals in the logistic analysis. All analyses were performed using the SAS statistical software package (version 9.4; SAS Institute, Cary, NC, USA).

## 3. Results

### 3.1. Baseline Characteristics

The most common cancer type was thyroid cancer, followed by gastric and colorectal cancer. Men were more likely to have a cancer diagnosis than women, except in cases of thyroid cancer. Most cancers were prevalent in those aged 70 or more (excluding thyroid, cervical, and breast cancer). The highest proportion of high-income (5Q) individuals was noted among men with prostate cancer (37.7%) (Table 1). As shown in Appendix A, the proportions of the population by income level were 28.1% (5Q), 22.0% (4Q), 17.8% (3Q), 14.7% (2Q), 14.3% (1Q), and 3% NMA beneficiaries. The total cancer incidence was 223,371 (men: 116,320, women: 107,051), with the income levels making up 29.5% (5Q), 20.4% (4Q), 16.0% (3Q), 13.5% (2Q), 15.6% (1Q), and 5% (NMA) of the total. Gastric cancer and breast cancer were the most prevalent cancers among men and women, respectively (Appendix A).

### 3.2. Incidence of Cancer and Disparities by Income

The age-standardized incidence rate (ASR) for all cancers was 496.6 per 100,000 in the population, as shown in Table 2 and Figure 1. The ASR of all cancers was the lowest in the highest income group (454.3 per 100,000), but the SII was not statistically significant (SII: −35.7, 95% CI: −93.65–22.15), and the RII was −0.07 (95% CI: −0.20–0.05). Colorectal, thyroid, cervical, and prostate cancer showed significant SIIs: colorectal cancer, −17.23 (95% CI: −31.64–−2.82); thyroid cancer, 15.97 (95% CI: 1.14–30.81); cervical cancer, −5.49 (95% CI: 6I.10–−4.87); and prostate cancer, 10.76 (95% CI: 0.29–21.23). Colorectal and cervical cancer had lower incidence rates in higher income groups, while thyroid and prostate cancer had higher incidence rates in higher income groups.

Among men, colorectal (SII: −29.11, 95% CI: −44.88–−13.34) and liver (SII: −27.48, 95% CI: −44.89–−1.06) cancer showed lower incidence rates with regard to higher income groups. Thyroid and prostate cancer showed higher incidence rates for higher income groups (Appendix A). Among women, colorectal (SII: −7.64, 95% CI: −15.19–−0.10) and cervical (SII: −7.77, 95% CI: −11.09–−4.44) cancers showed lower incidence rates for higher income groups, while thyroid cancer had higher incidence rates with respect to higher income groups (Appendix A).

### 3.3. Association between Income and Distant Stage at Diagnosis

The ORs for a distant stage at diagnosis for all cancers were greater for lower income groups (OR: 1.31, 95% CI: 1.26–1.36 in 1Q and 1.28, 1.23–1.33 in 2Q) than for 5Q among health insurance subscribers (Table 3 and Figure 2). The NMA group had an even higher OR for a distant stage at diagnosis (OR: 1.35, 95% CI: 1.16–1.56). Lower income groups, including the NMA group, had higher ORs for a distant stage for stomach and cervical cancers than those of 5Q. Lower income groups (1Q and 2Q) also had higher ORs for a distant stage at diagnosis for colorectal, liver, lung, and breast cancer, whereas the NMA group did not have significant ORs for a distant stage at diagnosis. Income level was not significantly associated with a distant stage at diagnosis for thyroid cancer.

## 4. Discussion

In this study, we used a large national database (KCCR linked with K-NHIS) consisting of all Korean citizens, making a representative study population. We found that income levels were associated with cancer incidence risk, and these associations differed by type of cancer. Disparities in colorectal and cervical cancer incidence favored higher income groups, while thyroid and prostate cancer showed disparities that were unfavorable to higher income groups. In addition, lower income was a significant predictor of a distant stage at diagnosis for all cancers.

In our study, the SII for colorectal cancer incidence was favorable for men with high incomes. Previous studies in several countries have produced inconsistent findings that vary in their health insurance systems and types of cancer covered by the national screening programs. Countries with universal health insurance coverage have demonstrated no significant association [14,15] between income level and colorectal cancer incidence or have found a lower incidence of colorectal cancer in low-income groups [16]. A study in the United States, where there is no universal health insurance coverage, reported that lower household income was associated with a higher incidence of colorectal cancer, mainly due to a low degree of accessibility to colorectal cancer screening (e.g., colonoscopy) [17]. Colonoscopy, sigmoidoscopy, and stool-based tests resulted in 45%, 34%, and 25% lower incidences of colorectal cancer relative to no screening. This is because screening tests enable the detection and removal of precancerous lesions, reducing the risk of developing colorectal cancer. In Korea, fecal immunochemical tests for colorectal cancer screening have been adopted by the NCSP since 2004 for individuals aged 50 years or older. A previous study in Korea reported that no significant inequality was observed in NSCP colorectal cancer screening, but the low-income group had a lower overall participation rate for colorectal cancer screening than the high-income group when private colorectal cancer screening rates were taken into account (51.7% vs. 62.0%) [18,19]. Thus, it is important to recognize the existing disparities in cancer incidence, with the exception of the NCSP for colorectal cancer screening, in addition to ensuring universal health insurance coverage, to reduce disparities in cancer incidence [20].

A higher cervical cancer incidence was noted in the low-income group in this study, consistent with previous findings [10,21]. In our study, the highest RII was noted, although the SII was relatively low due to low cancer incidence. In a meta-analysis, the low-income group had a higher OR (1.79, 95% CI: 1.62–1.97) for cervical cancer incidence than the high-income group [21]. Although infection with human papillomavirus is recognized as the primary risk factor for cervical cancer, multifactorial causes, including socioeconomic status, can contribute to human papilloma virus infection and subsequent cancer incidence [22]. Low income levels and low educational attainments are independent predictors of cervical cancer. Other risk factors, such as multiple sexual partners, an early age at first sexual practice, high parity, smoking, and alcohol consumption, could increase socioeconomic disparities in cervical cancer incidence. In addition, low accessibility to resources for prevention, such as human papillomavirus immunization, might aggravate such disparities [23].

We found that thyroid and prostate cancer had higher incidence rates in higher income groups. In Korea, the incidence of thyroid cancer has steeply increased due to overdiagnosis and increased screening for thyroid cancer [24]. Individuals with higher incomes may be at risk of overdiagnosis, as they can afford to pay for more sensitive medical tests than those with lower incomes [25]. Conversely, the higher prevalence of smoking and alcohol consumption in low-income groups might have a protective effect against the development of thyroid cancer [26]. The incidence of prostate cancer grew as income increased, consistent with previous publications [27,28]. This might reflect the use of prostate-specific antigen (PSA) testing in private cancer screening programs [29]. In Korea, the NCSP focuses on major cancers (stomach, breast, colorectal, cervical, and liver). However, individuals with higher incomes voluntarily participate in private comprehensive medical health checkups including thyroid ultrasonography, PSA, prostate ultrasonography, etc. Thus, disparities in thyroid and prostate cancer could reflect higher rates of detection of these kinds of cancers in higher income groups.

In this study, the SIIs of stomach [30], lung [31], and liver [32] cancer showed negative values, which is consistent with the findings of previous studies, in spite of statistical non-significance. The higher risk of incidence of stomach, lung, and liver cancer in the low-income group might be due, in part, to having more risk factors (*H. pylori* infection, dietary habits, smoking, and alcohol consumption) than those in the high-income group. Meanwhile, the SII of breast cancer was a positive value, which is explained by differences in reproductive factors, such as low parity and the use of hormone replacement therapy [33].

Lower income was a significant predictor of having a distant stage at diagnosis for cancers, with the exception of thyroid cancers, most of which are papillary thyroid microcarcinomas in Korea [34]. There are several possible explanations for this finding. First, the higher probability of an advanced stage at cancer diagnosis in low-income groups might partly reflect a low participation rate in cancer screening tests in low-income groups [35]. While the NCSP costs little or nothing, the barrier to participate in the NCSP includes possible costs for additional testing, difficulties in taking time off work, and the limited availability of healthcare facilities in rural areas [36]. Low levels of health literacy could negatively influence decisions to have cancer screening in low-income groups [37]. Second, the relatively low accessibility of medical services could be related to a delay from cancer development to cancer diagnosis. In addition, a lower perception of the significance of symptoms such as weight loss, dysphagia, and bleeding can lead to late-stage cancer diagnoses among low-income groups [38]. Greater socioeconomic deprivation was associated with a delay in symptomatic presentation, leading to an advanced stage at diagnosis [38]. Third, differences in lifestyles across income levels could play a role in cancer stage disparity at diagnosis [39]. The differences in health risk behaviors (smoking, obesity, and physical inactivity) and health environments according to socioeconomic status can contribute to different characteristics of cancer. Further research should address effective interventions to improve health literacy in low-income groups [40].

Several limitations must be considered. First, we only included insurance premiums (a proxy for income) as an indicator of socioeconomic status. Due to a lack of information, we could not use other indicators of socioeconomic status, such as education levels and having an occupation, in spite of their importance [16]. Second, our results may be of limited application to other countries due to differences in epidemiological characteristics and healthcare systems.

## 5. Conclusions

In conclusion, disparities in cancer incidence differed by cancer type. Colorectal and cervical cancer had higher incidence rates for individuals in the low-income group, while thyroid and prostate cancer showed higher incidence rates for those in the high-income group. Lower income was a significant predictor of a distant stage at diagnosis for cancers in general. This study emphasizes the need for further research into the underlying causes of disparities in cancer incidence and the stage at diagnosis as well as effective interventions that focus on enhancing health literacy in terms of cancer prevention to mitigate these disparities.

## Figures and Tables

**Figure 1 cancers-15-05898-f001:**
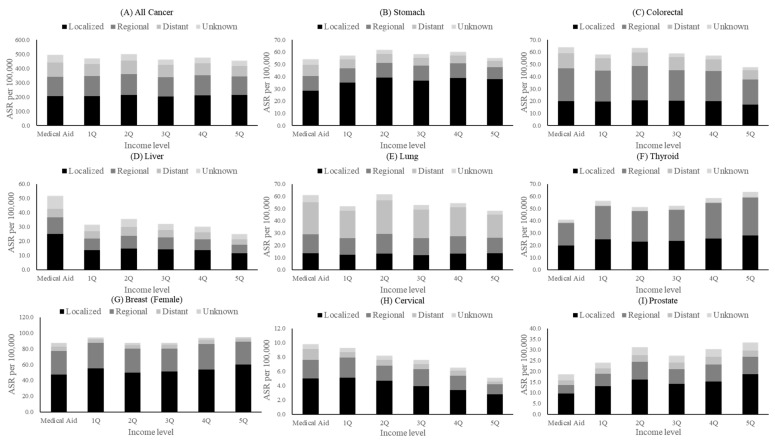
Age-standardized cancer incidence rates by cancer type and income level (2018) (per 100,000).

**Figure 2 cancers-15-05898-f002:**
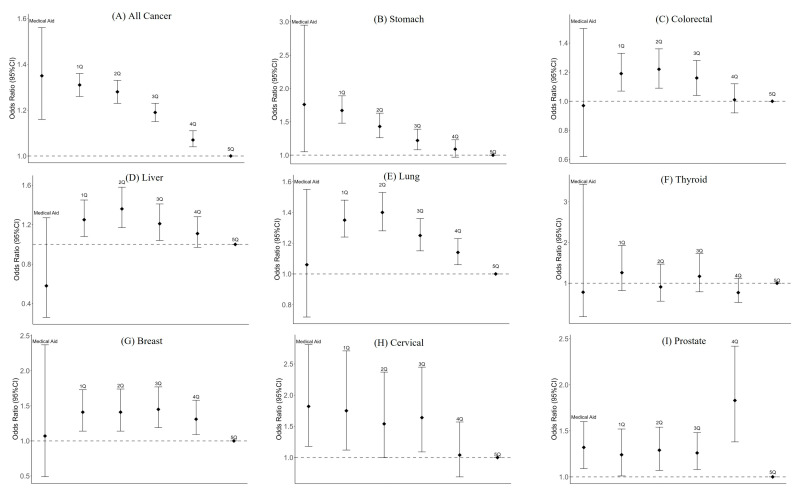
Adjusted odds ratios for distant-stage diagnosis for cancer across income levels. The estimates are adjusted for age, sex, employment type, residence, body mass index, smoking status, and alcohol consumption.

**Table 1 cancers-15-05898-t001:** Baseline characteristics of cancer patients by cancer type (*N* (%)).

Characteristics	Cancer Patients (All Cancers)	Gastric	Colorectal	Liver	Lung	Thyroid	Breast (Female)	Cervical (Female)	Prostate (Male)
Sex									
Male	104,069 (51.8)	17,528 (68.3)	15,439 (59.6)	10,616 (74.9)	16,503 (68.0)	6506 (23.6)	NA	NA	13,417 (100.0)
Female	97,027 (48.2)	8152 (31.7)	10,470 (40.4)	3559 (25.1)	7752 (32.0)	21,035 (76.4)	22,209 (100.0)	3295 (100.0)	NA
Age group									
20–29	4468 (2.2)	54 (0.2)	113 (0.4)	43 (0.3)	35 (0.1)	2081 (7.6)	130 (0.6)	71 (2.2)	2 (0.0)
30–39	11,242 (5.6)	495 (1.9)	622 (2.4)	229 (1.6)	175 (0.7)	5675 (20.6)	1724 (7.8)	576 (17.5)	5 (0.0)
40–49	25,867 (12.9)	2271 (8.8)	2010 (7.8)	1143 (8.1)	814 (3.4)	7505 (27.3)	7096 (32.0)	820 (24.9)	78 (0.6)
50–59	42,317 (21.0)	5749 (22.4)	5287 (20.4)	3420 (24.1)	3251 (13.4)	6964 (25.3)	6849 (30.8)	783 (23.8)	1091 (8.1)
60–69	49,008 (24.4)	7525 (29.3)	6731 (26.0)	4011 (28.3)	6919 (28.5)	3684 (13.4)	3983 (17.9)	490 (14.9)	4309 (32.1)
≥70	68,194 (33.9)	9586 (37.3)	11,146 (43.0)	5329 (37.6)	13,061 (53.8)	1632 (5.9)	2427 (10.9)	555 (16.8)	7932 (59.1)
Income									
Medicaid	9754 (4.9)	1176 (4.6)	1583 (6.1)	1118 (7.9)	1688 (7.0)	564 (2.0)	738 (3.3)	166 (5.0)	540 (4.0)
1Q	31,440 (15.6)	3993 (15.5)	4254 (16.4)	2311 (16.3)	3786 (15.6)	4026 (14.6)	3964 (17.8)	650 (19.7)	1753 (13.1)
2Q	27,225 (13.5)	3428 (13.3)	3595 (13.9)	2045 (14.4)	3135 (12.9)	3894 (14.1)	3297 (14.8)	570 (17.3)	1536 (11.4)
3Q	31,443 (15.6)	4054 (15.8)	4166 (16.1)	2242 (15.8)	3529 (14.5)	4657 (16.9)	3516 (15.8)	579 (17.6)	1821 (13.6)
4Q	41,340 (20.6)	5298 (20.6)	5162 (19.9)	2748 (19.4)	4758 (19.6)	6357 (23.1)	4486 (20.2)	666 (20.2)	2684 (20.0)
5Q	59,894 (29.8)	7731 (30.1)	7149 (27.6)	3711 (26.2)	7359 (30.3)	8043 (29.2)	6208 (28.0)	664 (20.2)	5083 (37.9)
Employment type									
Employee	58,356 (29.0)	7682 (29.9)	8047 (31.1)	4631 (32.7)	7193 (29.7)	6433 (23.4)	6553 (29.5)	1118 (33.9)	3794 (28.3)
Local	132,986 (66.1)	16,822 (65.5)	16,279 (62.8)	8426 (59.4)	15,374 (63.4)	20,544 (74.6)	14,918 (67.2)	2011 (61.0)	9083 (67.7)
Medicaid	9754 (4.9)	1176 (4.6)	1583 (6.1)	1118 (7.9)	1688 (7.0)	564 (2.0)	738 (3.3)	166 (5.0)	540 (4.0)
BMI									
<18.5	6700 (3.3)	870 (3.4)	842 (3.2)	459 (3.2)	822 (3.4)	944 (3.4)	741 (3.3)	99 (3.0)	460 (3.4)
18.5–25	120,740 (60.0)	15,426 (60.1)	15,661 (60.4)	8357 (59.0)	14,557 (60.0)	16,574 (60.2)	13,273 (59.8)	2036 (61.8)	8013 (59.7)
25–30	63,520 (31.6)	8040 (31.3)	8137 (31.4)	4626 (32.6)	7670 (31.6)	8663 (31.5)	7045 (31.7)	988 (30.0)	4242 (31.6)
≥30	10,136 (5.0)	1344 (5.2)	1269 (4.9)	733 (5.2)	1206 (5.0)	1360 (4.9)	1150 (5.2)	172 (5.2)	702 (5.2)
Smoking status									
Never	138,892 (69.1)	17,688 (68.9)	17,979 (69.4)	9775 (69.0)	16,782 (69.2)	18,971 (68.9)	15,347 (69.1)	2297 (69.7)	9344 (69.6)
Past	39,823 (19.8)	5112 (19.9)	5105 (19.7)	2782 (19.6)	4776 (19.7)	5479 (19.9)	4416 (19.9)	643 (19.5)	2595 (19.3)
Current	22,381 (11.1)	2880 (11.2)	2825 (10.9)	1618 (11.4)	2697 (11.1)	3091 (11.2)	2446 (11.0)	355 (10.8)	1478 (11.0)
Alcohol consumption									
No	118,335 (58.8)	15,032 (58.5)	15,434 (59.6)	8344 (58.9)	14,315 (59.0)	16,168 (58.7)	12,999 (58.5)	1918 (58.2)	7897 (58.9)
Yes	82,761 (41.2)	10,648 (41.5)	10,475 (40.4)	5831 (41.1)	9940 (41.0)	11,373 (41.3)	9210 (41.5)	1377 (41.8)	5520 (41.1)

NA: not applicable.

**Table 2 cancers-15-05898-t002:** Slope index of inequality (SII) and relative index of inequality (RII) of age-standardized incidence rate (per 100,000 in the population).

Cancer Type	Medical Aid	Health Insurance Subscribers					SII	SII 95% CI		RII	RII 95% CI	
		1Q	2Q	3Q	4Q	5Q						
All cancers	496.6	470.8	499.8	463.2	477.4	454.3	−35.75	−93.65	22.15	−0.07	−0.20	0.05
Stomach	54.4	57.1	62.0	58.6	60.3	55.4	−3.29	−14.38	7.79	−0.06	−0.25	0.13
Colorectal	64.0	58.0	63.7	59.2	57.1	47.8	−17.23	−31.64	−2.82	−0.30	−0.54	−0.05
Liver	51.8	31.6	35.7	32.2	30.2	25.1	−14.43	−29.94	1.09	−0.42	−0.87	0.03
Lung	61.3	52.0	61.7	53.0	54.4	48.2	−10.55	−27.10	5.99	−0.19	−0.49	0.11
Thyroid	41.0	56.3	51.2	52.4	58.7	63.6	15.97	1.14	30.81	0.30	0.02	0.57
Breast (female)	44.9	54.1	46.7	43.8	44.3	45.3	5.96	−7.93	19.84	0.07	−0.09	0.22
Cervical	9.8	9.3	8.2	7.5	6.6	5.0	−5.49	−6.10	−4.87	−0.71	−0.79	−0.63
Prostate	18.6	24.1	31.4	27.4	30.4	33.5	10.76	0.29	21.23	0.39	0.01	0.77

**Table 3 cancers-15-05898-t003:** Adjusted odds ratios (95% CIs) of distant-stage diagnosis for cancer across income groups.

Type of Cancer	Medical Aid	1Q	2Q	3Q	4Q	5Q
All cancers	1.35	1.31	1.28	1.19	1.07	Reference
	(1.16–1.56)	(1.26–1.36)	(1.23–1.33)	(1.15–1.23)	(1.04–1.11)	
Stomach	1.76	1.67	1.43	1.22	1.09	Reference
	(1.05–2.95)	(1.48–1.89)	(1.26–1.63)	(1.08–1.39)	(0.97–1.23)	
Colorectal	0.97	1.19	1.22	1.16	1.01	Reference
	(0.62–1.50)	(1.07–1.33)	(1.09–1.36)	(1.04–1.28)	(0.92–1.12)	
Liver	0.58	1.25	1.36	1.21	1.11	Reference
	(0.26–1.27)	(1.08–1.45)	(1.17–1.58)	(1.04–1.41)	(0.97–1.28)	
Lung	1.06	1.35	1.4	1.25	1.14	Reference
	(0.72–1.55)	(1.24–1.48)	(1.28–1.53)	(1.15–1.36)	(1.06–1.23)	
Thyroid	0.78	1.26	0.91	1.17	0.77	Reference
	(0.18–3.43)	(0.82–1.93)	(0.56–1.47)	(0.79–1.73)	(0.53–1.12)	
Breast	1.07	1.41	1.41	1.45	1.31	Reference
	(0.49–2.37)	(1.14–1.73)	(1.14–1.74)	(1.19–1.77)	(1.09–1.58)	
Cervical	1.82	1.75	1.54	1.64	1.04	Reference
	(1.18–2.81)	(1.12–2.71)	(1.00–2.37)	(1.09–2.45)	(0.69–1.57)	
Prostate	1.32	1.24	1.29	1.26	1.83	Reference
	(1.09–1.60)	(1.01–1.52)	(1.07–1.54)	(1.08–1.48)	(1.38–2.42)	

The estimates are adjusted for age, sex, employment type, residence, body mass index, smoking status, and alcohol consumption.

## Data Availability

Data supporting the findings of this study are available from the Korean NHIS and were used under license for this study (http://nhiss.nhis.or.kr, accessed on 1 September 2023). Restrictions apply to their availability (the data are not publicly available). The data are available upon request with permission from the Korean NHIS.

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
