# Peer review of "Disparities in Cancer Incidence across Income Levels in South Korea"

_cancers, 2023, doi:10.3390/cancers15245898_

Round 1

Reviewer 1 Report

Comments and Suggestions for Authors

This study utilized a comprehensive national Korean database to examine income-related disparities in cancer incidence and stage at diagnosis. Analyzing data from 223,361 cancer cases in 2018, the study found that lower income was associated with higher cancer incidence for certain types, like stomach and cervical cancers, and a greater likelihood of distant stage diagnosis across most cancers. Conversely, higher income groups showed increased incidence of thyroid and prostate cancers. These disparities were quantified using indices like the Slope Index of Inequality (SII). The findings highlight the need for targeted interventions and further research into the underlying causes of these disparities in cancer incidence and diagnosis stages, emphasizing the influence of socioeconomic factors on cancer outcomes.

Comments:

Abstract: The phrase "database with 2018 data." can be improved for clarity and precision. A more grammatically appropriate and clear version could be: "This study utilized data from a national cancer registry, specifically focusing on cases recorded in the year 2018..."

Line 57: The term "levels of units" is vague and could be confusing to readers unfamiliar with Korean administrative divisions. It's important to clarify that 'Sido' refers to administrative regions in Korea.

Line 68: The term "frequency analysis" is somewhat vague and could be interpreted in multiple ways. It's important to specify what exactly was analyzed using frequency analysis.

Line 86-88: Breaking it down into simpler components can enhance its comprehensibility.

Line 127: For enhanced clarity and ease of understanding, a rephrasing of this section is recommended, particularly the segment mentioning the NMA group.

Line 203: This statement may oversimplify the complex relationship between income, screening practices, and cancer diagnosis stage.

Line 225: The term "unfavorable" appears excessively and could lead to confusion. Consider using "higher incidence rates" as a clearer alternative.

Figure1-2: enlarging the font size for the labels on the x and y axes to improve readability.

Comments on the Quality of English Language

To improve clarity and comprehensibility, it is advised to rephrase some of the longer sentences. This approach can make the text more accessible and easier to follow.

Reviewer 2 Report

Comments and Suggestions for Authors

In section 1 (introduction), 1st paragraph for the 4th line needs citation.

Comments on the Quality of English Language

Copy editing may be required.

Author Response

#1. In section 1 (introduction), 1st paragraph for the 4th line needs citation.

Response: Thank you for highlighting this issue; we have added a citation to the first paragraph on the fourth line.

Reviewer 3 Report

Comments and Suggestions for Authors

thank you for allowing me to review this article assessing the impact of socioeconomic inequalities on cancer incidence and stage at diagnosis. the article is well written; the judgement criteria explicit. the completeness of the data is remarkable, given the use of a national registry covering the entire population. 

however, at no point does the percentage of missing data per item appear. should we deduce from this that no missing data existed? 

Conversely, if there were missing data, it would be useful to specify this and use multiple imputation as a statistical method, in order to clarify the influence of this data on the final results. 

the authors suggest a significant association between socio-economic deprivation and the over-incidence of certain cancers. how do the authors propose to remedy these results? 

in this registry, are there any data on time to diagnosis or time to care in relation to the date of diagnosis? these data could support the authors' diagnostic hypotheses, in particular inequalities in access to specialized care centers. 

Round 2

Reviewer 3 Report

Comments and Suggestions for Authors

the authors have responded point by point to questions and comments designed to improve the quality of the manuscript.

Author Response

Thank you for the reviewer's positive comments.